# Isolation and characterization of bis(silylene)-stabilized antimony(I) and bismuth(I) cations

Xuyang Wang[1,3], Binglin Lei[1,3], Zhaoyin Zhang[2,3], Ming Chen[1], Hua Rong[1], Haibin Song[1], Lili Zhao ⦾[2] ✉ & Zhenbo Mo ⦾[1] ✉

Monovalent group 15 cations $L_2Pn+$ (L = σ-donor ligands, Pn = N, P, As, Sb, Bi) have attracted significant experimental and theoretical interest because of their unusual electronic structures and growing synthetic potential. Herein, we describe the synthesis of a family of antimony(I) and bismuth(I) cations supported by a bis(silylene) ligand $[(TBDSi_2)Pn][BAr^F_4]$ (TBD = 1, 8, 10, 9-triazaboradecalin; $Ar^F$ = 3,5-$CF_3$-$C_6H_3$; Pn = Sb, (**2**); Bi, (**3**)). The structures of **2** and **3** have been unambiguously characterized spectroscopically and by X-ray diffraction analysis and DFT calculations. They feature bis-coordinated Sb and Bi atoms which exhibit two lone pairs of electrons. The reactions of **2** and **3** with methyl trifluoromethane sulfonate provide a approach for the preparation of dicationic antimony(III) and bismuth(III) methyl complexes. Compounds **2** and **3** serve as 2e donors to group 6 metals (Cr, Mo), giving rise to ionic antimony and bismuth metal carbonyl complexes **6**–**9**.

Over the past decades, low-valent main group chemistry constantly gives surprising discoveries. Among them, the monoatomic zerovalent group 14 compounds $L_2E$ (E = C, Si, Ge, Sn, Pb; L = σ-donor ligands), termed as tetrylones, have moved from theoretical curiosities to synthetically accessible molecules by using a base-stabilization protocol[1–3]. These species have unique electronic structures and display synthetic potential as soluble molecular allotropes[4]. In spite of this, the chemistry of tetrylone homologs still contains much uncertainty[5]. Cationic group 15 element compounds are valuable main group species owing to their isoelectronic relationship with the corresponding group 14 element compounds shown in Fig. 1 and their wide implications in catalysis[6–22]. Recently, group 15 cations of the type $R_2Pn^+$ (Pn = P, As, Sb, Bi; R = aryl groups), cationic homologs of carbene, were described by Beckmann et al.[23–26]. Monovalent group 15 cations $L_2Pn^+$ (Pn = N, P, As, Sb, Bi) are isoelectronic analogs of tetrylones, which are highly sought-after synthetic targets[5]. Similar to the E(0) atoms, the $Pn(I)^+$ atoms possess two lone pairs of electrons and $Pn(I)^+$ cations of the lighter elements (N, P, As)[27–30] are most familiar.

Their heavier cationic counterparts ($Sb(I)^+$ and $Bi(I)^+$) have been much less explored in spite of significant progress that has been recorded in recent years in the isolation of neutral antimony(I) and bismuth(I) complexes[31–41]. In a seminal publication in 2020, Roesky et al. reported the isolation and characterization of Sb(I) and Bi(I) cations stabilized by a cyclic (alkyl)(amino)carbene (CAAC)[42], but study on their reactivity remains elusive. The bis(diisopropylamino)cyclopropenylidene-stabilized Sb(I) cation was prepared by Stephan et al., but was found to decompose at room temperature[43]. More recently, Majumdar et al. obtained a bis(phosphine)-coordinated Sb(I) cation and disclosed its fascinating coordination behavior with transition metals[44]. Therefore, the investigation of antimony(I) and bismuth(I) cations is still in its infancy and their reactivity is waiting to be explored.

The effective stabilization of antimony(I) and bismuth(I) cations relies heavily on the appropriate supporting ligands. N-heterocyclic silylenes, homologs of N-heterocyclic carbenes, have been shown to be capable of stabilizing low-valent main group complexes as a result of their strong σ-donating ability and partial π-accepting ability[45–55]. Of

[1]State Key Laboratory and Institute of Elemento-Organic Chemistry, Frontiers Science Center for New Organic Matter, College of Chemistry, Nankai University, 94 Weijin Road, 300071 Tianjin, China. [2]Institute of Advanced Synthesis, School of Chemistry and Molecular Engineering, State Key Laboratory of Materials-Oriented Chemical Engineering, Nanjing Tech University, 211816 Nanjing, China. [3]These authors contributed equally: Xuyang Wang, Binglin Lei, Zhaoyin Zhang. ✉e-mail: ias_llzhao@njtech.edu.cn; zhenbo.mo@nankai.edu.cn

particular note are the recent reports by Driess and co-workers that the stannylone and plumbylone[52,53], which are isoelectronic with antimony(I) and bismuth(I) cations, can be prepared with a bis(NHSi) ligand. However, antimony and bismuth complexes with NHSi ligands have remained unknown to date. Herein, we report the synthesis, and characterization of NHSi-stabilized antimony(I) and bismuth(I) cations [(TBDSi$_2$)Pn][BAr$^F_4$] (TBD = 1, 8, 10, 9-triazaboradecalin; Ar$^F$ = 3,5-CF$_3$−C$_6$H$_3$; Pn = Sb, (**2**); Bi, (**3**)). Reactions of compounds **2** and **3** with MeOTf have led to methyl antimony(III) and bismuth(III) dications (**4, 5**). The coordination of **2** and **3** towards group 6 metals (Cr, Mo) affords a series of ionic antimony and bismuth metal carbonyl complexes [(TBDSi$_2$)Pn → M(CO)$_5$][BAr$^F_4$] (Pn = Sb, M = Cr, **6**; Pn = Sb, M = Mo, **7**; Pn = Bi, M = Cr, **8**; Pn = Bi, M = Mo, **9**).

## Results

### Preparation and characterization of 1

First, we synthesized a bidentate NHSi ligand by lithiation of 1,8,10,9-triazaboradecalin (TBD), a cyclic triaminoborane[56], with 2 equiv. of n-BuLi followed by salt-metathesis with 2 equiv. of the chlorosilylene [PhC(N$^t$Bu)$_2$]SiCl (Fig. 2a). The rigid TBD spacer was selected to prevent the dimerization of two NHSi moieties and generate a suitable space for metal coordination. The donation of the nitrogen lone pairs into the vacant p-orbital on boron would impact the electronic property of the bridged NHSis. Recent studies have revealed that the nature of the spacers is of crucial importance to the success of bidentate NHSis[57–64]. The $^{29}$Si NMR peaks of **1** (δ = −22.3 ppm) are shifted upfield from those of the bis(NHSi)ferrocene (43.3 ppm)[57], bis(NHSi) xanthene

(17.3 ppm)[58], bis(NHSi) carborane (18.9 ppm)[45], and bis(NHSi)aniline (−12.9 ppm)[59] ligands (Fig. 2b), indicating its enhanced electron-donating properties. The $^{11}$B NMR spectrum of **1** shows a broad signal at 27.2 ppm. The structure of **1** was characterized by single-crystal X-ray diffraction analysis, and its molecular structure is shown in Supplementary Fig. 60. The Si atom features a trigonal-pyramidal geometry as indicated by the 272.1° sum of the bond angles around the silicon atom. The Si...Si separation in **1** is 3.005(2) Å which is shorter than the Si···Si bond in bis(NHSi)carborane (3.267 Å) and bis(NHSi) xanthene (4.316 ppm), but longer than that of the bis(NHSi)aniline (2.9 Å). The Si1−N1 bond distance in **1** (1.761(5) Å) is comparable to the value of 1.781(1) in bis(NHSi)aniline. The N−B bond distances of 1.452 and 1.443 Å are comparable to those in B(NMe$_2$)$_3$ (1.44 Å), indicating significant N−B double-bond character.

### Isolation and characterization of 2 and 3

Treatment of **1** with a variety of antimony(III) and bismuth(III) halides such as SbCl$_3$, SbBr$_3$, BiCl$_3$, BiBr$_3$, [IPr → SbBr$_3$][64] or [IPr → BiBr$_3$][64] led to elemental Sb and Bi and intractable mixtures. Reaction of **1**, [IPr→SbBr$_3$], with Na[BAr$^F_4$] in THF at −30 °C followed by reduction with 2 equiv. of potassium graphite (KC$_8$), however, afforded the desired antimony(I) cation [(TBDSi$_2$)Sb][BAr$^F_4$] (**2**) as a yellow solid in 53% yield (Fig. 3). A similar reaction using [IPr → BiBr$_3$] gave a red-brown solution, from which the bismuth(I) cation [(TBDSi$_2$)Bi][BAr$^F_4$] (**3**) was isolated as orange crystals in 44% yield (Fig. 4). Complexes **2** and **3** are stable both in the solid state and in solution under an inert atmosphere. They have been characterized by NMR spectroscopy, UV/Vis-NIR spectroscopy, single-crystal X-ray diffraction, and elemental analysis. The $^1$H NMR spectra of **2** and **3** show one sharp singlet at 1.30 and 1.26 ppm, respectively, for the tert-butyl groups, consistent with their C2v symmetry. The $^{29}$Si NMR resonance of **2** (−8.71 ppm) is shifted downfield relative to the signal from **3** (−28.6 ppm). The UV−Vis spectrum of **2** in THF shows two absorption bands at 244 and 306 nm, whereas that of **3** has two broad absorption bands at 240 and 305 nm. These values are close to the predicted absorptions for the anionic parts of **2** and **3** (**2**: 238 and 329 nm; **3**: 227 and 350 nm) which were calculated using the TDDFT-PBE0 method with the def2-svp basis set. The maximum bands (329 and 350 nm) are assigned to the HOMO → LUMO-1 transition.

**Fig. 1 | Isoelectronic relationship of monoatomic zerovalent group 14 compounds and monovalent group 15 cations.** Monovalent group 15 cations L$_2$Pn+ (Pn = N, P, As, Sb, Bi) are isoelectronic analogs of tetrylones L$_2$E (E = C, Si, Ge, Sn, Pb).

**Fig. 2 | Synthetic routes to the bis(silylene) ligand and compare with bis(silylene)s with different spacers. a** Synthetic routes to the bis(silylene) ligand (1) with 1,8,10,9-triazaboradecalin; **b** bis(silylene)s with different spacers.

## Single crystals of 2 and 3

Single crystals of **2** and **3** were obtained by slow diffusion of hexane into the saturated THF solutions of **2** and **3**, respectively, and their molecular structures are depicted in Fig. 4. Both compounds feature a nearly planar six-membered $BN_2Si_2Pn$ ring in which the sum of the internal bond angles is 719.1° (Sb) and 719.0° (Bi), respectively. The central Sb and Bi atoms of **2** and **3** are both bis-coordinated with the silicon atoms of NHSi, which are well separated from the weakly coordinating counteranion $[BAr^F_4]^-$. The Si–Sb–Si bond angle of **2** (85.33(2)°) is comparable to the P–Sb–P bond angle in bis(phosphine)-coordinated Sb(I) cation (89.53(3)°), but is much smaller than the C–Sb–C bond angle in CAAC-stabilized Sb(I) cation (111.87(7)°). The Sb–Si bond lengths are 2.4619(9) and 2.455(5) Å and are shorter than the sum of the covalent radii of the Sb and Si (2.56 Å), but longer than the Sb=Si bond length in $[TripSbSi(SiMe^tBu_2)_2]$ (Trip = 2,4,6-$^i$Pr-$C_6H_2$, 2.4146(7) Å)[65]. The Bi–Si bond lengths of 2.5610(8) and 2.5565(9) Å are markedly shorter with respect to those in $[Bi\{Si(SiMe_3)_3\}_2][Li(THF)_4]$ (2.668(5) and 2.672(4) Å)[66].

## Quantum chemical calculations

To probe the electronic structures of **2** and **3**, density functional theoretical (DFT) calculations were performed on the anionic parts of **2** and **3** at the BP86+D3(BJ)/def2-TZVPP level. The optimized structures are in good agreement with the X-ray-derived structures, and both have a singlet electronic ground state, which is 47.0 and 36.8 kcal mol⁻¹ lower in energy than their triplet state isomers, respectively (Supplementary Fig. 68). The Wiberg bond orders of the Sb–Si and Bi–Si bonds are 1.22 and 1.18, respectively, suggesting their partial double-bond character. The large bond dissociation energies for the scission of the Pn–Si bonds (233.6 kcal mol⁻¹ for **2**; 214.7 kcal mol⁻¹ for **3**) unambiguously support their strong bonding nature. The molecular orbitals defining the lone pairs of electrons located at the central Sb and Bi atoms are displayed in Fig. 5a and Supplementary Fig. 69. The highest occupied molecular orbital (HOMO) of **2** and **3** represents a π-type lone pair. The HOMO-5 of **2** and HOMO-6 of **3** are dominated by a σ-type lone pair. The natural population analysis (NPA) charges in **2** (Sb:

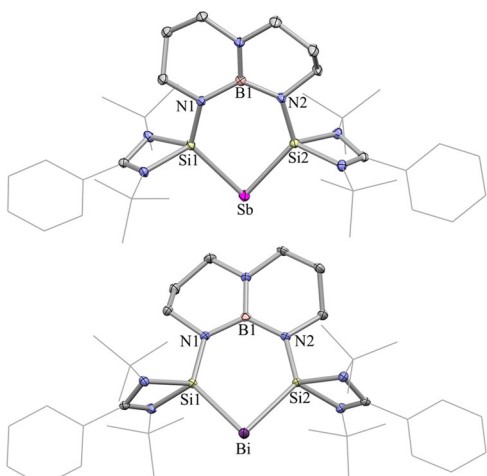

**Fig. 4 | The molecular structures of 2 and 3.** The molecular structures of the antimony(I) and bismuth(I) cations **2** and **3** showing 30% probability of ellipsoids. The anion $[BAr^F_4]$ and hydrogen atoms are omitted for clarity. Selected bond distances (Å) and angles (°): for **2**: Sb1–Si1 2.4619(6), Sb1–Si2 2.455(5), Si1–N1 1.703(3), Si2–N2 1.706(2), Si1–Sb1–Si2 85.33(3)°, N1–B1–N2 123.1(2)°; for **3**: Bi1–Si1 2.561(8), Bi1–Si2 2.557(1), Si1–N1 1.710(3), Si2–N2 1.712(2), Si1–Bi1–Si2 82.10(3)°, N1–B1–N2 123.5(3)°.

**Fig. 3 | Synthesis of the antimony(I) and bismuth(I) cations $[(TBDSi_2)Pn][BAr^F_4]$ (Pn = Sb, 2; Bi, 3).** Reaction of **1**, [IPr→SbBr₃] or [IPr→BiBr₃], with Na[BAr^F_4] in THF at −30 °C followed by reduction with 2 equiv. of potassium graphite (KC₈).

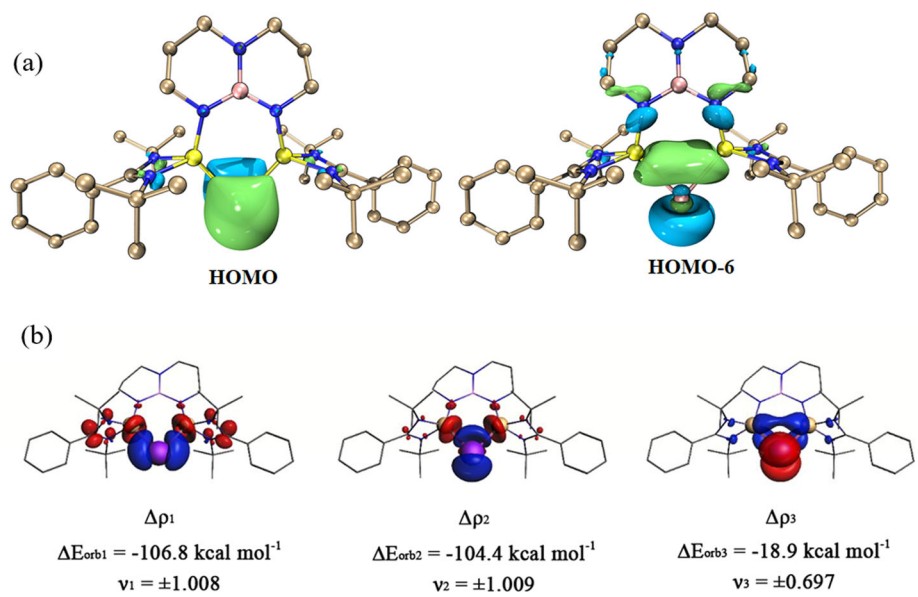

**Fig. 5 | Molecular orbital analysis and plot of deformation densities Δρ1–Δρ3 of 3. a** Selected frontier orbitals of **3**, determined by DFT calculations (isovalue = 0.05). **b** Plot of deformation densities Δρ1–Δρ3 of **3** by using the Bi⁺ and Bis(silylene) interacting fragments in their Singlet (S) states, with the associated interaction energies $\Delta E_{orb}$ (in kcal mol⁻¹). The eigenvalues $\nu$ are a measure of the relative amount of charge transfer. The direction of the charge flow is from red → blue.

HOMO

HOMO-6

Δρ1
$\Delta E_{orb1}$ = -106.8 kcal mol⁻¹
$\nu_1 = \pm1.008$

Δρ2
$\Delta E_{orb2}$ = -104.4 kcal mol⁻¹
$\nu_2 = \pm1.009$

Δρ3
$\Delta E_{orb3}$ = -18.9 kcal mol⁻¹
$\nu_3 = \pm0.697$

−0.340; Si: +1.608) and **3** (Bi: −0.275; Si: +1.587) indicate strong electron donation from the Si atoms toward the formal empty molecular orbitals on Sb and Bi.

More detailed information about the Pn−Si bonds in **2** and **3** is available from the state-of-the-art energy decomposition analysis with natural orbitals for chemical valence (EDA-NOCV)[67,68] method by considering the interacting fragments in various electronic states. Details about the method can be found in Supplementary Information. As shown in Table 1, the most appropriate fragments for the two Bi−Si bonds in **3** are Bi$^+$ and Bis(silylene) species in their electronic singlet (S) states, because the fragments that give the smallest orbital term are the best choice for a faithful representation of the bond[69,70]. The largest contribution to the attractive interactions in **3** comes from the orbital term $\Delta E_{orb}$, which provides 49.7% of the total attraction. The electrostatic attraction $\Delta E_{elstat}$ is 46.1%, which is slightly weaker than the orbital term $\Delta E_{orb}$. Note that the weak dispersion forces provide the remaining 4.3% of the total attraction, which is weak but still not negligible.

**Table 1 | EDA-NOCV results of 2, 3 by using the Pn$^+$ (Pn = Bi, Sb) and Bis(silylene) interacting fragments in their Singlet (S) states at the BP86+D3(BJ)/TZ2P-ZORA//BP86+D3(BJ)/def2-TZVPP level of theory**

| Fragments | Bi$^+$ (S, $6s^26p_x^06p_y^06p_z^2$) +Bis(silylene) (S) | Sb$^+$ (S, $5s^25p_x^05p_y^05p_z^2$) +Bis(silylene) (S) |
|---|---|---|
| $\Delta E_{int}$ | −239.1 | −261.0 |
| $\Delta E_{Pauli}$ | 277.7 | 295.8 |
| $\Delta E_{disp}$[a] | −22.0(4.3%) | −20.7(3.7%) |
| $\Delta E_{elstat}$[a] | −238.1(46.1%) | −239.4(43.0%) |
| $\Delta E_{orb}$[a] | −256.7(49.7%) | −296.6(53.3%) |
| $\Delta E_{orb1}$[b] | −106.8(41.6%) | −125.4(42.3%) |
| $\Delta E_{orb2}$[b] | −104.4(40.7%) | −121.4(40.9%) |
| $\Delta E_{orb3}$[b] | −18.9(7.4%) | −20.7(7.0%) |
| $\Delta E_{rest}$[b] | −26.6(10.3%) | −29.1(9.8%) |

Energy values are given in kcal mol$^{-1}$.
[a]The values in parentheses give the percentage contribution to the total attractive interactions $\Delta E_{elstat} + \Delta E_{orb} + \Delta E_{disp}$.
[b]The values in parentheses give the percentage contribution to the total orbital interactions $\Delta E_{orb}$.

Table 1 shows that there are three major orbital interactions (i.e., $\Delta E_{orb1}$–$\Delta E_{orb3}$) which provide 89.7% of $\Delta E_{orb}$. Figure 5b shows that the strongest two orbital interactions $\Delta E_{orb1}$ and $\Delta E_{orb2}$ are mainly due to the formation of two Si→Bi dative bonds, which are denoted as Bis(silylene) (HOMO, HOMO-1)→Bi$^+$ (LUMO, LUMO + 1) σ-donation. The third orbital interaction, $\Delta E_{orb3}$ (−8.9 kcal mol$^{-1}$), is mainly due to the π backdonation from the occupied $p$ orbital of Bi$^+$ (HOMO) to the antibonding orbital (LUMO + 7) of the Bis(silylene) ligand. The left contribution of orbital relaxation $\Delta E_{rest}$ of 10.3% originates from the polarization within the fragments. Inspection of the composition of the interacting orbitals is given in Supplementary Fig. 70. The similar bonding nature of two Sb−Si bonds in **2** can be found in Table 1 and Supplementary Figs. 71, 72. The EDA-NOCV analysis clearly showed that the structure of **2** and **3** can be most accurately described as >Si:→:Pn$^+$:←:Si<, denoting NHSi-stabilized antimony(I) and bismuth(I) cations.

## Reactivity studies

The reactivity of **2** and **3** towards methyl trifluoromethane sulfonate (MeOTf) was investigated in order to examine their nucleophilic nature. Addition of 1.0 equiv. of MeOTf to **2** in fluorobenzene followed by anion exchange with Na[BAr$^F_4$] immediately afforded a colorless solution. After workup the methyl antimony(III) dication [(TBDSi$_2$)SbMe][BAr$^F_4$]$_2$ (**4**) was obtained in 78% yield (Fig. 6a). The bismuth(III) dication [(TBDSi$_2$)BiMe][BAr$^F_4$]$_2$ (**5**) was prepared in 71% yield using the same synthetic protocol. The $^1$H NMR spectra of **4** and **5** in d$_8$-THF reveal a singlet at 2.05 and 2.30 ppm, respectively for the methyl groups, which is shifted downfield compared to those of the parent compounds SbMe$_3$ and BiMe$_3$, due to their electron-deficient nature. Single-crystal X-ray diffraction analysis of **5** revealed that the bismuth center adopts a trigonal-pyramidal coordination geometry (Fig. 6b). The Si–Bi–Si angle of the chelating ligand (81.70°) is comparable to that of **3** (82.113°). The Si–Bi distances (2.651 and 2.636 Å) are significantly elongated compared to those in **3** because the oxidation of **3** weakens the electron delocalization between Bi and Si atoms. The Bi−C bond length of 2.300 Å is akin to those in BiMe$_3$ (on average: 2.26 Å) and the dimethylbismuth cation [BiMe$_2$(SbF$_6$)] (2.22 Å)[71]. Compound **5** crystallizes with a severely disordered anion [BAr$^F_4$] and consequently its metric parameters are not discussed here. The remaining lone pairs located on Sb and Bi atoms can be found in the HOMO−6 and HOMO−8

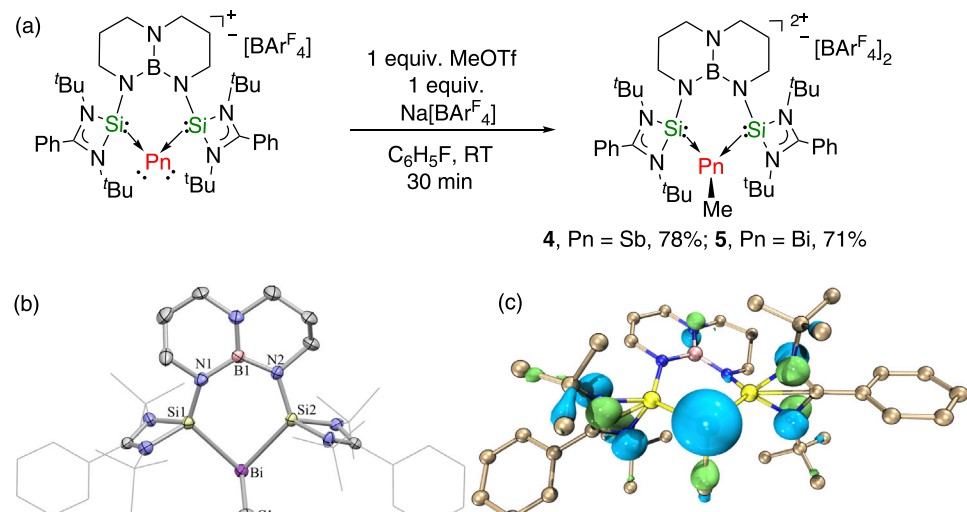

**Fig. 6 | Synthetic routes to the complexes 4–5 and the molecular structure of 5. a** Synthetic routes to the methyl antimony(III) and bismuth(III) dications [(TBDSi$_2$)PnMe][BAr$^F_4$]$_2$ (Pn = Sb, **4**; Bi, **5**). **b** The molecular structure of the methyl bismuth(III) dication **5** showing 30% probability ellipsoids. The anion [BAr$^F_4$], solvent, and hydrogen atoms are omitted for clarity. Selected bond distances (Å) and angles (°): Bi1–Si1 2.651, Bi1–Si2 2.636, Bi1–C1 2.300, Si1–N1 1.681(8), Si2–N2 1.69(1), Si1–Bi1–Si2 81.70°, Si1–Bi1–C1 98.76°, Si2–Bi1–C1 95.55°, N1–B1–N2 124(1)°. **c** HOMO-8 of **5**, determined by DFT calculations (isovalue = 0.05).

**Fig. 7 | Synthetic routes to complexes 6–9.** The reactions of **2** and **3** with in situ prepared [M(CO)$_5$(thf)] (M = Cr, Mo) were taken in a 1:1 ratio in fluorobenzene at room temperature.

(see Fig. 6c and Supplementary Fig. 73). NBO analysis reveals that the lone pairs feature a high s-character (73% for **4** and 83% for **5**). NPA revealed positive charges of +0.45 and +0.53 at Sb and Bi atoms, respectively.

In order to evaluate the Lewis acidity of compounds **2–5**, we calculated the fluoride ion affinity (FIA), and hydride ion affinity (HIA) of compounds **2–5** which is detailed in Supplementary Table 4. The calculated FIA and HIA of **2** (114.2 and 113.0 kcal mol$^{-1}$, respectively) and **3** (115.3 and 113.7 kcal mol$^{-1}$, respectively), are much lower than those of **4** (192.8 and 201.4 kcal mol$^{-1}$, respectively) and **5** (190.2 and 196.4 kcal mol$^{-1}$, respectively). The calculated values for **4** and **5** are comparable to that of bismuth(III) dication with a hydro tris (pyrazolyl)borate ligand (234 kcal mol$^{-1}$)[17]. The Lewis acidity of **2–5** was further estimated by Gutmann− Beckett method using Et$_3$PO as an internal standard. When OPEt$_3$ was mixed with one equivalent of **2**, **3**, **4**, or **5** in CD$_2$Cl$_2$, the peak shifted to $\delta$ 50.8, 51.3, 54.0, and 55.2 ppm (Supplementary Table 5), respectively, indicating a weak coordination of Et$_3$P=O to them that might be due to the strong coordination of two silylene moieties to the Sb and Bi centers.

**Coordination chemistry of 2 and 3 towards transition metal carbonyls**

Ligands based on heavy main group elements have attracted considerable attention on account of their unique structural and electronic properties as well as their synthetic potential. Antimony(I) and bismuth(I) cations should be a class of interesting ligands for the coordination of transition metals as the presence of two lone pairs on the Bi(I) and Sb(I) centers. However, there is only one report on the coordination behavior of bis(phosphine)-stabilized antimony(I) cation[44]. The reactions of **2** and **3** with in situ prepared [M(CO)$_5$(thf)] (M = Cr, Mo) taken in 1:1 ratio in fluorobenzene result in the formation of ionic antimony and bismuth group 6 carbonyl complexes [((TBDSi$_2$)Pn → M(CO)$_5$][BAr$^F_4$] (Fig. 7, Pn = Sb, M = Cr, **6**; Pn = Sb, M = Mo, **7**; Pn = Bi, M = Cr, **8**; Pn = Bi, M = Mo, **9**). Complexes **6–9** were isolated in good yields (70–80%) as orange crystals and fully characterized by NMR spectroscopy, IR spectroscopy, elemental analysis, and single-crystal X-ray diffraction. The $^1$H NMR spectra of **6–9** recorded in CDCl$_3$ show one resonance for the $^t$Bu groups, revealing the symmetry of their structure in solution. Their $^{13}$C NMR spectra contained two signals above 200 ppm for the equatorial or axial carbonyl groups. The $^{29}$Si NMR resonances of **6** (−15.59 ppm) and **8** (−81.46 ppm) are shifted upfield relative to those of **2** (−8.71 ppm) and **3** (−28.60 ppm). The IR spectra of complexes **6–9** show several bands in the region 2063–1901 cm$^{-1}$, which are comparable to those of the neutral N,C,N-chelated pnictinidene group 6 carbonyl complexes[72].

Single crystals of **6–9** suitable for X-ray diffraction were produced by slow diffusion of hexane to the saturated fluorobenzene solutions at room temperature. The molecular structures of **6–9** closely resemble each other (Fig. 8). Complexes **6** and **7** crystallize in the monoclinic space group P2$_1$/c and complexes **8** and **9** crystallize in the triclinic space group P-1. The Sb and Bi atoms in complexes **6–9** are three-coordinate with a distorted trigonal-pyramidal geometry (ΣSb = 325.47° (**6**), 325.45 (**7**), ΣBi = 314.53 (**8**)), suggesting the presence of another lone pair of electrons. The Sb and Bi atoms are shifted out of the plane defined by Si1−Si2−Pn. The Pn−M bond distances in **6–9** (2.702(1), 2.8584 (6), 2.802(3), and 2.9450(7) Å, respectively) are comparable to those in N,C,N-chelated pnictinidene group 6 carbonyl complexes (2.7482(6), 2.8916(6), 2.8144(19) and 2.9425(5) Å), but are significantly longer than those in the reported Sb(III) complex [Me$_3$Pn → Cr(CO)$_5$][73] (Pn = Sb, 2.6171(3) Å; Pn = Bi, 2.7049(4) Å) and [Ph$_3$Pn → Mo(CO)$_5$][74] (Pn = Sb, 2.7560(17) Å; Pn = Bi, 2.8327(10) Å). The Pn–Si distances are elongated compared to those in **2** and **3** due to the decreased electron delocalization between Pn and Si atoms. In complexes **6–9**, the M−C$_{axial}$ bond distances are shorter than those of M−C$_{equatorial}$ bond as the Sb(I) and Bi(I) donors enhanced π-back donation to the axial carbonyl group.

Density functional theory (DFT) calculations were conducted to disclose the electronic structure, stability, and bonding nature of **6** and **8**. The highest occupied molecular orbitals (HOMO) of **6** and **8** consist mainly with the lone pairs at Pn (Pn = Sb and Bi) atoms (Supplementary Fig. 74). The HOMO-2 represents the orbital overlap between the p-type lone pair on Pn atoms and the vacant d orbital on Cr atom. We investigated the nature of the Pn−Cr bonds in **6** and **8** by using the EDA-NOCV method. As detailed in Table 2, the interacting fragments in **8** should be better described as Bi$^+$[bis(silylene)] and Cr(CO)$_5$ in their Singlet (S) states. The relative size of orbital ($\Delta E_{orb}$) and electrostatic ($\Delta E_{elstat}$) energies reveals that the interaction is somewhat more electrostatic than covalent in nature, while the dispersion contribution is also quite stronger and not negligible, contributing 19.0% of the total attraction. There are three major orbital interactions (i.e., $\Delta E_{orb1}$–$\Delta E_{orb3}$) and the relative deformation densities (i.e., $\Delta \rho_1$–$\Delta \rho_3$), which are detailed in Supplementary Fig. 75. The strongest orbital interaction (i.e., $\Delta E_{orb1}$ = −43.7 kcal mol$^{-1}$) arises mainly from the formation of the Bi → Cr dative bond. The second and third orbital interactions $\Delta E_{orb2}$ and $\Delta E_{orb3}$ (−3.5 and −3.1 kcal mol$^{-1}$, respectively), are mainly attributable to the π backdonation from the Cr center to the Bi center. Detailed information about the composition of the interacting orbitals is given in Supplementary Fig. 76. Similar bonding nature of Sb −Cr bond in **6** can be found in Table 2 and Supplementary Figs. 77, 78.

## Discussion

In summary, this work reports that the bis(silylene)-supported antimony(I) and bismuth(I) cations can be synthesized through a one-pot reaction of bis(NHSi) ligand (**1**), [IPr → PnBr$_3$] (Pn = Sb, Bi), with Na[Bar$^F_4$] in THF at −30 °C followed by reduction with 2 equiv. of KC$_8$. X-ray diffraction analysis and DFT calculations reveal the presence of two-coordinated Sb and Bi centers, each featuring two lone pairs of electrons. A preliminary study of the reactivity showed that the

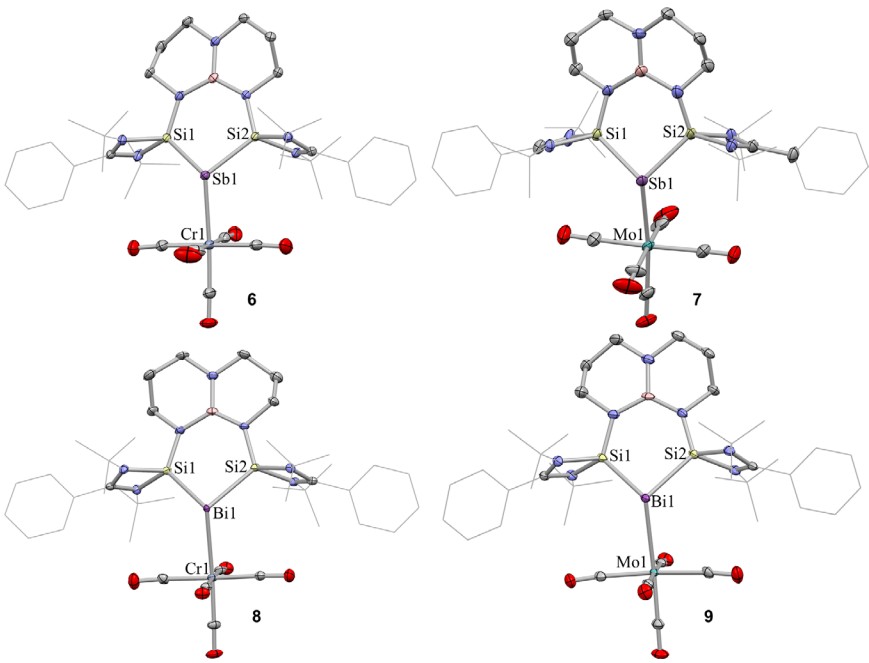

**Fig. 8 | The molecular structures of 6–9.** The molecular structure of the **6–9** showing a 30% probability of ellipsoids. The anion [BAr$^F_4$], solvent, and hydrogen atoms are omitted for clarity. Selected bond distances (Å) and angles (°): for **6**: Sb1–Si1 2.4950(9), Sb1–Si2 2.498(1), Sb1–Cr1 2.702(1), Si1–Sb1–Si2 88.02(3)°; for **7**:

Sb1–Si1 2.485(1), Sb1–Si2 2.498(1), Sb1–Mo1 2.8584(6), Si1–Sb1–Si2 87.76(4)°; for **8**: Bi1–Si1 2.609(3), Bi1–Si2 2.588(4), Bi1–Cr1 2.802(3), Si1–Bi1–Si2 84.0(1)°; for **9**: Bi1–Si1 2.600(2), Bi1–Si2 2.599(1), Bi1–Mo1 2.9450(7), Si1–Bi1–Si2 83.69(6)°.

reaction of **2** and **3** with 1 equiv. of MeOTf afforded the examples of dicationic antimony(III) and bismuth(III) methyl complexes (**4** and **5**). The coordination of **2** and **3** towards group 6 metals (Cr, Mo) gives ionic antimony and bismuth metal carbonyl complexes **6–9**, demonstrating the nucleophilic property of the antimony(I) and bismuth(I) cations. Further applications of bis(silylene)-supported cationic antimony and bismuth complexes are the subjects of ongoing work.

## Methods
### General procedures
All reactions were carried out under a dry and oxygen-free argon or dinitrogen atmosphere by using Schlenk techniques or under an argon or dinitrogen atmosphere in a Vigor glovebox. The argon or dinitrogen

## Table 2 | EDA-NOCV results of 6, 8 by using the Pn⁺[bis(silylene)] (Pn = Bi, Sb) and Cr(CO)₅ interacting fragments in their Singlet (S) states at the BP86 + D3(BJ)/TZ2P-ZORA//BP86 + D3(BJ)/def2-TZVPP level of theory

| Fragments | Bi⁺[bis(silylene)](S) +Cr(CO)₅(S) | Sb⁺[bis(silylene)] (S) +Cr(CO)₅ (S) |
|---|---|---|
| $\Delta E_{int}$ | −57.4 | −57.5 |
| $\Delta E_{Pauli}$ | 93.1 | 95.2 |
| $\Delta E_{disp}$[a] | −28.6(19.0%) | −28.9(18.9%) |
| $\Delta E_{elstat}$[a] | −63.9(42.4%) | −65.9(43.2%) |
| $\Delta E_{orb}$[a] | −58.1(38.6%) | −57.9(37.9%) |
| $\Delta E_{orb1}$[b] | −43.7(75.2%) | −42.1(72.7%) |
| $\Delta E_{orb2}$[b] | −3.5(6.0%) | −4.2(7.2%) |
| $\Delta E_{orb3}$[b] | −3.1(5.3%) | −3.4(5.9%) |
| $\Delta E_{rest}$[b] | −7.9(13.6%) | −8.2(14.2%) |

Energy values are given in kcal mol⁻¹.
[a]The values in parentheses give the percentage contribution to the total attractive interactions $\Delta E_{elstat}$ + $\Delta E_{orb}$ + $\Delta E_{disp}$.
[b]The values in parentheses give the percentage contribution to the total orbital interactions $\Delta E_{orb}$.

in the glovebox was constantly circulated through a copper/molecular sieves catalyst unit. The oxygen and moisture concentrations in the glovebox atmosphere were monitored by an O₂/H₂O Combi-Analyzer (Vigor LG2400/750TS-F) to ensure both were always below 1 ppm. Full information on the synthetic, spectroscopic, crystallographic, and computational methods is given in Supplementary Information.

**Preparation of Compound 1.** n-BuLi (5 mL of a 1.6 M solution in hexane, 8.0 mmol) was added dropwise to a solution of 1,8,10,9-triaminoborane (556 mg, 4.0 mmol) in THF (20 mL) at −30 °C. The mixture was warmed to room temperature and stirred for a further 16 h. The resulting white solution was cooled to −30 °C, and then a solution of [PhC(N$^t$Bu)₂]SiCl (2.36 g, 8.0 mmol) in THF was added slowly. The color of the mixture changed from colorless to dark red gradually. After warming to room temperature and stirring for 4 h, the solvent was removed in a vacuum and the residue was extracted with hexane (30 mL). The extract was concentrated to 5 mL and crystallized overnight at −30 °C to afford compound **1** as a red solid (1.84 g, 2.8 mmol, 70%). Single crystals suitable for X-ray diffraction studies were obtained by recrystallization from a hexane solution at room temperature.

¹H NMR (400 MHz, C₆D₆, 298 K): δ 7.15–7.13 (m, 2H, Ar-*H*), 7.04–7.02 (m, 2H, Ar-*H*), 6.97–6.95 (m, 4H, Ar-*H*), 6.90–6.86 (m, 2H, Ar-*H*), 3.25 (m, 4H, NC*H₂*), 3.18 (t, 4H, NC*H₂*, J = 6.2 Hz), 2.07 (m, 4H, CH₂-C*H₂*-CH₂), 1.39 (s, 36H, C(C*H₃*)₃).

¹³C NMR (101 MHz, C₆D₆, 298 K): δ 158.1 (s, N*C*N), 135.9 (s, Ar*C*), 130.7 (s, Ar*C*), 128.9 (s, Ar*C*), 128.8 (s, Ar*C*), 127.6 (s, Ar*C*), 127.5 (s, Ar*C*), 53.2 (s, *C*(CH₃)₃), 50.0 (s, N*C*H₂), 40.2 (s, N*C*H₂), 32.7 (s, C(*C*H₃)₃), 30.0 (s, CH₂-*C*H₂-CH₂).

¹¹B NMR (128 MHz, C₆D₆, 298 K): δ 27.2 ppm (br).

²⁹Si NMR (79 MHz, C₆D₆, 298 K): δ −22.3 ppm (s).

Anal. calcd. for C₃₆H₅₈BN₇Si₂: C, 65.93; H, 8.91; N, 14.95. Found: C, 65.65; H, 8.86; N, 14.60.

**Preparation of Compound 2.** THF (15 mL) was cooled to −30 °C and added to a mixture of compound **1** (200 mg, 0.30 mmol), (IPr)SbBr₃

(225 mg, 0.30 mmol), $KC_8$ (81 mg, 0.60 mmol) and $Na[BAr^F_4]$ (266 mg, 0.30 mmol). Then the mixture was warmed to room temperature and stirred for 3 h and filtered. The solvent was removed in a vacuum, and the residue was washed with hexane (8 mL), diethyl ether (8 mL), and toluene (8 mL) to yield compound **2** as a yellow powder (265 mg, 0.16 mmol, 53%). Single crystals suitable for X-ray diffraction studies were obtained by slow diffusion of hexane into the saturated THF solutions of **2** at room temperature.

$^1H$ NMR (400 MHz, THF-$d_8$, 298 K): δ 7.79 (s, 8H, Barf-Ar-*H*), 7.65–7.55 (m, 14H, Ar-*H*), 3.28 (br, 4H, NC*H*$_2$), 3.02 (t, 4H, NC*H*$_2$, *J* = 6.9 Hz), 1.95 (br, 4H, CH$_2$-C*H*$_2$-CH$_2$), 1.30 (s, 36H, C(C*H*$_3$)$_3$).

$^{13}C$ NMR (101 MHz, THF-$d_8$, 298 K): δ 176.8 (s, N*C*N), 162.8 (q, $J_{C-B}$ = 51 Hz, Barf-Ar-*C*), 135.6 (s, Ar*C*), 132.4 (s, Ar*C*), 130.3 (s, Ar*C*), 130.2 (m, Barf-Ar-*C*), 129.9 (m, Barf-Ar-*C*), 129.5 (s, Ar*C*), 129.4 (s, Ar*C*), 128.4 (s, Ar*C*), 125.5 (q, $J_{C-F}$ = 274 Hz, Barf-*C*F$_3$), 118.2 (m, Barf-Ar-*C*), 56.7 (s, N*C*(CH$_3$)$_3$), 50.0 (s, N*C*H$_2$), 42.9 (s, N*C*H$_2$), 31.2 (s, C(*C*H$_3$)$_3$), 28.2 (s, *C*H$_2$-*C*H$_2$-*C*H$_2$).

$^{11}B$ NMR (128 MHz, THF-$d_8$, 298 K): δ 26.6 ppm (br), −6.5 ppm (s, Barf-*B*).

$^{19}F$ NMR (377 MHz, THF-$d_8$, 298 K): δ −63.4 ppm (s).

$^{29}Si$ NMR (79 MHz, THF-$d_8$, 298 K): δ −8.7 ppm (s).

Anal. calcd. for $C_{68}H_{70}B_2SbF_{24}N_7Si_2$: C, 49.78; H, 4.30; N, 5.98. Found: C, 49.42; H, 4.51; N, 5.58.

**Preparation of Compound 3.** THF (15 mL) was cooled to −30 °C and added to a mixture of compound **1** (200 mg, 0.30 mmol), (IPr)BiBr$_3$ (251 mg, 0.30 mmol), $KC_8$ (81 mg, 0.60 mmol) and $Na[BAr^F_4]$ (266 mg, 0.30 mmol). Then the mixture was warmed to room temperature and stirred for 3 h and filtered. The solvent was removed in a vacuum, and the residue was washed with hexane (8 mL), diethyl ether (8 mL), and toluene (8 mL) to yield compound **3** as an orange powder (231 mg, 0.13 mmol, 44%). Single crystals suitable for X-ray diffraction studies were obtained by liquid phase diffusion of a solution of THF with hexane at room temperature.

$^1H$ NMR (400 MHz, CDCl$_3$, 298 K): δ 7.71 (s, 8H, Barf-Ar-*H*), 7.59–7.47 (m, 9H, Ar-*H*), 7.39–7.32 (m, 5H, Ar-*H*), 3.18 (br, 4H, NC*H*$_2$), 2.94 (br, 4H, NC*H*$_2$), 1.91 (br, 4H, CH$_2$-C*H*$_2$-CH$_2$), 1.26 (s, 36H, C(C*H*$_3$)$_3$).

$^{13}C$ NMR (101 MHz, CDCl$_3$, 298 K): δ 175.1 (s, N*C*N), 161.9 (q, $J_{C-B}$ = 51 Hz, Barf-Ar-*C*), 135.0 (s, Ar*C*), 131.7 (s, Ar*C*), 130.4 (s, Ar*C*), 129.4 (s, Ar*C*), 129.2 (m, Barf-Ar-*C*), 128.8 (m, Barf-Ar-*C*), 128.7 (s, Ar*C*), 127.5 (s, Ar*C*), 124.7 (q, $J_{C-F}$ = 273 Hz, Barf-*C*F$_3$), 117.6 (m, Barf-Ar-*C*), 55.8 (s, N*C*(CH$_3$)$_3$), 49.1 (s, N*C*H$_2$), 42.6 (s, N*C*H$_2$), 31.6 (s, C(*C*H$_3$)$_3$), 27.5 (s, *C*H$_2$-*C*H$_2$-*C*H$_2$).

$^{11}B$ NMR (128 MHz, THF-$d_8$, 298 K): δ 27.0 ppm (br), −6.5 ppm (s, Barf-*B*).

$^{19}F$ NMR (377 MHz, THF-$d_8$, 298 K): δ −65.2 ppm (s).

$^{29}Si$ NMR (79 MHz, THF-$d_8$, 298 K): δ −28.6 ppm (s).

Anal. calcd. for $C_{68}H_{70}B_2BiF_{24}N_7Si_2$: C, 47.26; H, 4.08; N, 5.67. Found: C, 47.55; H, 4.21; N, 5.10.

**Preparation of Compound 4.** MeOTf (10.3 μL, 0.098 mmol) was added to **2** (160 mg, 0.098 mmol) in fluorobenzene (10 mL) at −30 °C. Then $Na[BAr^F_4]$ (87 mg, 0.098 mmol) was added to this mixture immediately. The solution was stirred for 30 min and the color changed from yellow to white. The reaction solution was diluted with 5 mL of hexane with rigorous stirring to give a white precipitate. After decanting the supernatant, the precipitate was washed with diethyl ether (6 mL) and dried under vacuum to give **4** as a colorless, moisture-sensitive solid (199 mg, 0.076 mmol, 78% yield).

$^1H$ NMR (400 MHz, THF-$d_8$, 298 K): δ 7.78 (s, 16H, Barf-Ar-*H*), 7.72–7.70 (m, 4H, Ar-*H*), 7.65–7.61 (m, 6H, Ar-*H*), 7.57 (s, 8H, Barf-Ar-*H*), 3.42 (br, 4H, NC*H*$_2$), 3.08 (t, 4H, NC*H*$_2$, *J* = 6.3 Hz), 2.09 (br, 4H, CH$_2$-C*H*$_2$-CH$_2$),1.99 (s, 3H, Sb(C*H*$_3$), 1.32 (s, 18H, C(C*H*$_3$)$_3$), 1.29 (s, 18H, C(C*H*$_3$)$_3$).

$^{13}C$ NMR (101 MHz, THF-$d_8$, 298 K): δ 181.5 (s, N*C*N), 162.6 (q, $J_{C-B}$ = 49 Hz, Barf-Ar-*C*), 135.4 (s, (s, Ar*C*), 133.2 (s, Ar*C*), 130.0 (m, Barf-Ar-*C*), 129.7 (m, Barf-Ar-*C*), 129.6(s, Ar*C*), 129.4 (s, Ar*C*), 129.3(s, Ar*C*), 129.2(s, Ar*C*), 129.1 (s, Ar*C*), 128.1(s, Ar*C*), 125.3 (q, $J_{C-F}$ = 274 Hz, Barf-*C*F$_3$), 118.0 (m, Barf-Ar-*C*), 57.5 (s, N*C*(CH$_3$)$_3$), 56.4 (s, N*C*(CH$_3$)$_3$), 49.7 (s, N*C*H$_2$), 43.9 (s, N*C*H$_2$), 31.0(s, C(*C*H$_3$)$_3$), 30.7 (s, C(*C*H$_3$)$_3$), 28.4 (s, Sb(*C*H$_3$)), 27.5 (s, *C*H$_2$-*C*H$_2$-*C*H$_2$).

$^{11}B$ NMR (128 MHz, THF-$d_8$, 298 K): δ 30.1 ppm (br), −6.5 ppm (s, Barf-*B*).

$^{19}F$ NMR (377 MHz, THF-$d_8$, 298 K): δ −63.3 ppm (s).

$^{29}Si$ NMR (79 MHz, THF-$d_8$, 298 K): δ −15.5 ppm (s).

Anal. calcd. for $C_{107}H_{90}B_3SbF_{49}N_7Si_2$: C, 49.14; H, 3.47; N, 3.75. Found: C, 48.89; H, 3.77; N, 3.34.

**Preparation of Compound 5.** MeOTf (9.7 μL, 0.092 mmol) was added to **3** (160 mg, 0.092 mmol) in fluorobenzene (10 mL) at −30 °C. Then $Na[BAr^F_4]$ (82 mg, 0.092 mmol) was added to this mixture immediately. The solution was stirred for 30 min and the color changed from yellow to white. The reaction solution was diluted with 5 mL of hexane with rigorous stirring to give a white precipitate. After decanting the supernatant, the precipitate was washed with diethyl ether (6 mL) and dried under vacuum to give **5** as a colorless, moisture sensitive solid (177 mg, 0.065 mmol, 71% yield). Single crystals suitable for X-ray diffraction studies were obtained by slow diffusion of hexane into the saturated fluorobenzene solutions of **5** at 5 °C.

$^1H$ NMR (400 MHz, THF-$d_8$, 298 K): δ 7.79 (s, 16H, Barf-Ar-*H*), 7.75–7.65 (m, 10H, Ar-*H*), 7.58 (s, 8H, Ar-*H*), 3.47 (t, 4H, NC*H*$_2$ *J* = 4.8 Hz), 3.09 (m, 4H, NC*H*$_2$), 2.25 (s, 3H, Bi(C*H*$_3$)), 2.09 (m, 4H, CH$_2$-C*H*$_2$-CH$_2$), 1.31 (s, 18H, C(C*H*$_3$)$_3$), 1.30 (s, 18H, C(C*H*$_3$)$_3$).

$^{13}C$ NMR (101 MHz, THF-$d_8$, 298 K): δ 179.7 (s, N*C*N), 162.6 (q, $J_{C-B}$ = 51 Hz, Barf-Ar-*C*), 135.4 (s, Ar*C*), 132.9 (s, Ar*C*), 130.2 (m, Barf-Ar-*C*), 129.8. (s, Ar*C*), 129.7 (m, Barf-Ar-*C*), 129.6. (s, Ar*C*), 129.5 (s, Ar*C*), 129.3 (s, Ar*C*), 128.9 (s, Ar*C*), 128.8 (s, Ar*C*), 125.3 (q, $J_{C-F}$ = 273 Hz, Barf-*C*F$_3$), 118.0 (m, Barf-Ar-*C*), 57.0 (s, N*C*(CH$_3$)$_3$), 56.0 (s, N*C*(CH$_3$)$_3$), 49.9 (s, N*C*H$_2$), 44.1 (s, N*C*H$_2$), 31.3(s, C(*C*H$_3$)$_3$), 30.6 (s, C(*C*H$_3$)$_3$), 28.4 (s, Bi(*C*H$_3$)), 27.8 (s, *C*H$_2$-*C*H$_2$-*C*H$_2$),

$^{11}B$ NMR (128 MHz, THF-$d_8$, 298 K): δ 29.3 ppm (br), −6.5 ppm (s, Barf-*B*).

$^{19}F$ NMR (377 MHz, THF-$d_8$, 298 K): δ −65.2 ppm (s).

$^{29}Si$ NMR (79 MHz, THF-$d_8$, 298 K): δ −31.2 ppm (s).

Anal. calcd. for $C_{107}H_{90}B_3BiF_{49}N_7Si_2$: C, 47.56; H, 3.36; N, 3.63. Found: C, 47.12; H, 3.70; N, 3.33.

**Preparation of Compound 6.** A sample of $[Cr(CO)_6]$ (13 mg, 0.06 mmol) in THF (6 mL) was irradiated by UV-lamp for 1 h to generate a yellow solution of $[Cr(CO)_5(thf)]$, that was then added to a solution of complex **2** (100 mg, 0.06 mmol) in fluorobenzene (5 mL) at room temperature The resulting yellow solution was stirred for a further 1 h at room temperature. The solvent was removed in a vacuum, and the residue was washed with hexane (10 mL) and extracted with Et$_2$O (5 mL). The solvent was evaporated to yield compound **6** as a yellow powder (88 mg, 0.048 mmol, 80%). Single crystals suitable for X-ray diffraction studies were obtained by slow diffusion of hexane to the saturated fluorobenzene solutions at RT.

$^1H$ NMR (400 MHz, CDCl$_3$, 298 K): δ 7.71 (s, 8H, Barf-Ar-*H*), 7.62–7.60 (m, 2H, Ar-*H*), 7.52–7.44 (m, 10H, Ar-*H*), 7.35–7.34 (m, 2H, Ar-*H*), 3.23 (br, 4H, NC*H*$_2$), 2.93 (br, 4H, NC*H*$_2$), 1.92 (br, 4H, CH$_2$-C*H*$_2$-CH$_2$), 1.27 (s, 36H, C(C*H*$_3$)$_3$).

$^{13}C$ NMR (101 MHz, CDCl$_3$, 298 K): δ 222.7 (s, *C*O-ax.), 219.1 (s, *C*O-eq.) δ 178.3 (s, N*C*N), 160.7 (q, $J_{C-B}$ = 51 Hz, Barf-Ar-*C*), 133.8 (s, Ar*C*), 131.2 (s, Ar*C*), 128.1 (s, Ar*C*), 127.8 (m, Barf-Ar-*C*), 127.7 (m, Barf-Ar-*C*), 127.6 (s, Ar*C*), 127.5 (s, Ar*C*), 127.4 (s, Ar*C*), 126.9 (s, Ar*C*), 125.7 (s, Ar*C*), 123.5 (q, $J_{C-F}$ = 274 Hz, Barf-*C*F$_3$), 116.5 (m, Barf-Ar-*C*), 55.2 (s, N*C*(CH$_3$)$_3$), 47.9 (s, N*C*H$_2$), 41.4 (s, N*C*H$_2$), 29.9. (s, C(*C*H$_3$)$_3$), 25.8 (s, *C*H$_2$-*C*H$_2$-*C*H$_2$).

[11]B NMR (128 MHz, CDCl$_3$, 298 K): δ 29.0 ppm (br), −6.6 ppm (s, Barf-*B*).

[19]F NMR (377 MHz, CDCl$_3$, 298 K): δ −62.4 ppm (s).

[29]Si NMR (79 MHz, CDCl$_3$, 298 K): δ −15.6 ppm (s).

IR (CO, cm$^{-1}$): 2045, 1967, 1930, and 1909 cm$^{-1}$.

Anal. calcd. for C$_{73}$H$_{70}$B$_2$CrF$_{24}$N$_7$O$_5$SbSi$_2$: C, 47.84; H, 3.85; N, 5.35. Found: C, 47.52; H, 3.60; N, 4.93.

**Preparation of Compound 7**. A sample of Mo[(CO)$_6$] (16 mg, 0.06 mmol) in THF (6 mL) was irradiated by UV-lamp for 1 h to generate a yellow solution of [Mo(CO)$_5$(thf)], that was then added to a solution of complex **2** (100 mg, 0.06 mmol) in fluorobenzene (5 mL) at room temperature The resulting yellow solution was stirred for a further 1 h at room temperature. The solvent was removed in a vacuum, and the residue was washed with hexane (10 mL) and extracted with Et$_2$O (5 mL). The solvent was evaporated to yield compound **7** as a yellow powder (81 mg, 0.043 mmol, 71%).

[1]H NMR (400 MHz, CDCl$_3$, 298 K): δ 7.70 (s, 8H, Barf-Ar-*H*), 7.63–7.60 (m, 2H, Ar-*H*), 7.56–7.45 (m, 10H, Ar-*H*), 7.38–7.33 (m, 2H, Ar-*H*), 3.23 (br, 4H, NC*H$_2$*), 2.93 (br, 4H, NC*H$_2$*), 1.92 (br, 4H, CH$_2$-C*H$_2$*-CH$_2$), 1.27 (s, 36H, C(C*H$_3$*)$_3$).

[13]C NMR (101 MHz, CDCl$_3$, 298 K): 210.5 (s, *C*O-ax.), 208.0 (s, *C*O-eq.) δ 178.9 (s, N*C*N), 161.7 (q, *J*$_{C-B}$ = 49 Hz, Barf-Ar-*C*), 134.8 (s, Ar*C*), 132.2 (s, Ar*C*), 129.0 (m, Barf-Ar-*C*), 128.7 (m, Barf-Ar-*C*), 128.6 (s, Ar*C*), 128.5 (s, Ar*C*), 128.4 (s, Ar*C*), 128.0 (s, Ar*C*), 126.5 (s, Ar*C*), 125.7 (s, Ar*C*), 124.5 (q, *J*$_{C-F}$ = 274 Hz, Barf-*C*F$_3$), 117.5 (m, Barf-Ar-*C*), 56.3 (s, N*C*(CH$_3$)$_3$), 48.9 (s, N*C*H$_2$), 42.4 (s, N*C*H$_2$), 30.9. (s, C(*C*H$_3$)$_3$), 26.8 (s, CH$_2$-*C*H$_2$-CH$_2$).

[11]B NMR (128 MHz, CDCl$_3$, 298 K): δ 29.3 ppm (br), −6.6 ppm (s, Barf-*B*).

[19]F NMR (377 MHz, CDCl$_3$, 298 K): δ −62.4 ppm (s).

[29]Si NMR (79 MHz, CDCl$_3$, 298 K): δ −15.1 ppm (s).

IR (CO, cm$^{-1}$): 2063, 1987, 1936, and 1901 cm$^{-1}$.

Anal. calcd. for C$_{73}$H$_{70}$B$_2$MoF$_{24}$N$_7$O$_5$SbSi$_2$: C, 46.72.; H, 3.76; N, 5.22. Found: C, 46.31; H, 3.15; N, 4.93.

**Preparation of Compound 8**. A sample of [Cr(CO)$_6$] (15 mg, 0.06 mmol) in THF (6 mL) was irradiated by UV-lamp for 1 h to generate a yellow solution of [Cr(CO)$_5$(thf)], that was then added to a solution of complex **3** (100 mg, 0.06 mmol) in fluorobenzene (5 mL) at room temperature The resulting yellow solution was stirred for a further 1 h at room temperature. The solvent was removed in a vacuum, and the residue was washed with hexane (10 mL) and extracted with Et$_2$O (5 mL). The solvent was evaporated to yield compound **8** as a yellow powder (81 mg, 0.040 mmol, 75%). Single crystals suitable for X-ray diffraction studies were obtained by slow diffusion of hexane to the saturated fluorobenzene solutions at RT.

[1]H NMR (400 MHz, CDCl$_3$, 298 K): δ 7.71 (s, 8H, Barf-Ar-*H*), 7.63–7.60 (m, 2H, Ar-*H*), 7.53–7.49 (m, 8H, Ar-*H*), 7.44–7.42 (m, 2H, Ar-*H*), 7.38–7.36 (m, 2H, Ar-*H*), 3.24 (br, 4H, NC*H$_2$*), 2.93 (br, 4H, NC*H$_2$*), 1.93 (br, 4H, CH$_2$-C*H$_2$*-CH$_2$), 1.26 (s, 36H, C(C*H$_3$*)$_3$).

[13]C NMR (101 MHz, CDCl$_3$, 298 K): 220.9 (s, *C*O-ax.), 220.8 (s, *C*O-eq.) δ 177.0 (s, N*C*N), 160.1 (q, *J*$_{C-B}$ = 49 Hz, Barf-Ar-*C*), 133.8 (s, Ar*C*), 131.1 (s, Ar*C*), 128.1 (s, Ar*C*), 128.0 (m, Barf-Ar-*C*), 127.8 (s, Ar*C*), 127.7 (m, Barf-Ar-*C*), 127.6 (s, Ar*C*), 127.4 (s, Ar*C*), 127.2 (s, Ar*C*), 126.2 (s, Ar*C*), 123.5 (q, *J*$_{C-F}$ = 274 Hz, Barf-*C*F$_3$), 116.5 (m, Barf-Ar-*C*), 55.0 (s, N*C*(CH$_3$)$_3$), 48.0 (s, N*C*H$_2$), 41.4 (s, N*C*H$_2$), 30.2. (s, C(*C*H$_3$)$_3$), 26.0 (s, CH$_2$-*C*H$_2$-CH$_2$).

[11]B NMR (128 MHz, CDCl$_3$, 298 K): δ 28.7 ppm (br), −6.6 ppm (s, Barf-*B*).

[19]F NMR (377 MHz, CDCl$_3$, 298 K): δ −62.4 ppm (s).

[29]Si NMR (79 MHz, CDCl$_3$, 298 K): δ −81.6 ppm (s).

IR (CO, cm$^{-1}$): 2037, 1963, 1928, and 1905 cm$^{-1}$.

Anal. calcd. for C$_{73}$H$_{70}$B$_2$CrF$_{24}$N$_7$O$_5$BiSi$_2$: C, 45.66; H, 3.67; N, 5.11.Found: C, 46.12; H, 3.40; N, 5.53.

**Preparation of Compound 9**. A sample of [Mo(CO)$_6$] (15 mg, 0.06 mmol) in THF (6 mL) was irradiated by UV-lamp for 1 h to generate a yellow solution of [Mo(CO)$_5$(thf)], that was then added to was added to a solution of complex **3** (100 mg, 0.06 mmol) in fluorobenzene (5 mL) at room temperature The resulting yellow solution was stirred for a further 1 h at room temperature. The solvent was removed in vacuum, and the residue was washed with hexane (10 mL) and extracted with Et$_2$O (5 mL). The solvent was evaporated to yield compound **9** as a yellow powder (81 mg, 0.040 mmol, 70%). Single crystals suitable for X-ray diffraction studies were obtained by slow diffusion of hexane to the saturated fluorobenzene solutions at RT.

[1]H NMR (400 MHz, CDCl$_3$, 298 K): δ 7.70 (s, 8H, Barf-Ar-*H*), 7.63–7.60 (m, 2H, Ar-*H*), 7.56–7.48 (m, 8H, Ar-*H*), 7.44–7.42 (m, 2H, Ar-*H*), 7.37–7.36 (m, 2H, Ar-*H*), 3.24 (br, 4H, NC*H$_2$*), 2.93 (br, 4H, NC*H$_2$*), 1.93 (br, 4H, CH$_2$-C*H$_2$*-CH$_2$), 1.26 (s, 36H, C(C*H$_3$*)$_3$).

[13]C NMR (101 MHz, CDCl$_3$, 298 K): 208.5 (s, *C*O-ax.), 200.0 (s, *C*O-eq.) δ 176.7 (s, N*C*N), 160.7 (q, *J*$_{C-B}$ = 51 Hz, Barf-Ar-*C*), 133.8 (s, Ar*C*), 131.1 (s, Ar*C*), 130.5 (s, Ar*C*), 128.3 (s, Ar*C*), 128.0 (m, Barf-Ar-*C*), 127.7 (m, Barf-Ar-*C*), 127.5 (s, Ar*C*), 127.4 (s, Ar*C*), 126.4 (s, Ar*C*), 126.0 (s, Ar*C*), 123.5 (q, *J*$_{C-F}$ = 274 Hz, Barf-*C*F$_3$), 116.4 (m, Barf-Ar-*C*), 54.5 (s, N*C*(CH$_3$)$_3$), 47.9 (s, N*C*H$_2$), 41.4 (s, N*C*H$_2$), 30.2. (s, C(*C*H$_3$)$_3$), 26.4 (s, CH$_2$-*C*H$_2$-CH$_2$).

[11]B NMR (128 MHz, CDCl$_3$, 298 K): δ 29.8 ppm (br), −6.6 ppm (s, Barf-*B*).

[19]F NMR (377 MHz, CDCl$_3$, 298 K): δ −62.4 ppm (s).

[29]Si NMR (79 MHz, CDCl$_3$, 298 K): δ −78.6 ppm (s).

IR (CO, cm$^{-1}$): 2057, 1958, 1944, and 1913 cm$^{-1}$.

Anal. calcd. for C$_{73}$H$_{70}$B$_2$MoF$_{24}$N$_7$O$_5$BiSi$_2$: C, 44.64; H, 3.59; N, 4.99. Found: C, 45.32; H, 3.23; N, 4.73.

## Data availability

All data generated or analyzed during this study are included in this manuscript (and its Supplementary Information). Details about materials and methods, experimental procedures, characterization data, and NMR spectra are available in the Supplementary Information. The structures of **1**−**9** in the solid state were determined by single-crystal X-ray diffraction studies and the crystallographic data have been deposited with the Cambridge Crystallographic Data Centre under nos. CCDC 2208058 (**1**), 2208059 (**2**), 2208060 (**3**), 2208061(**5**), 2245419 (**6**), 2245420 (**7**), 2245422 (**8**), and 2245421 (**9**). Copies of the data can be obtained free of charge on application to CCDC. These data can be obtained free of charge from The Cambridge Crystallographic Data Centre via www.ccdc.cam.ac.uk/data request/cif. All data are also available from corresponding authors upon request.

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

## Acknowledgements

We thank the financial support from the National Natural Science Foundation of China (Nos. 22071124, Z.M.; 22188101, Z.M.; 21973044, L.Z.), Frontiers Science Center for New Organic Matter at Nankai University (C029215001), the Fundamental Research Funds for the Central Universities (No. 63223010, Z.M.), Nankai University, Young Elite Scientists Sponsorship Program by Tianjin, the Natural Science Foundation of the Jiangsu province [No. BK20211587, L.Z.], Nanjing Tech University [Nos. 39837123, L.Z.; and 39837132, L.Z.], and the International Cooperation Project at Nanjing Tech University. L.Z. appreciates the high-performance center of Nanjing Tech University for supporting the computational resources. We are sincerely grateful to Professor Qi-Lin Zhou for his generous support.

## Author contributions

Z.M. designed the project. X.W. and B.L. carried out the experiments. Z.Z. did the theoretical calculations. L.Z. and Z.M. analyzed the data and wrote the manuscript. M.C., H.R., and H.S. performed single-crystal X-ray diffraction studies.

## Competing interests

The authors declare no competing interests.
