## [Peer Review File · Nature Communications]

Isolation and Characterization of Bis(silylene)-Stabilized Antimony(I) and Bismuth(I) CationsReviewers' Comments:

Reviewer #1:

Remarks to the Author:

In their manuscript, Mo and co-workers describe the preparation and complete characterization (including X-ray diffraction analysis) of novel As(I) and Sb(I) cations stabilized by a bidentate bis(silylene) ligand. This study follows the previous achievement by Roesky, Stephan and others on strongly related species and is in line with recent computational studies (mainly by Frenking and coworkers) on group 13-15 analogues of tetrylones. In addition, the reactivity of the title compounds was briefly explored in the reaction with MeOTf which leads to the corresponding dicationic species. DFT calculations were in addition carried out to mainly support the occurrence of two lone-pairs in the cations.

In my opinion, this is a solid piece of work that has been competently and carefully carried out. As the manuscript is in general well-written and the literature up-to-date, I am glad to support its acceptance in NatCommun. The following minor issues might be addressed in a revision:

(1) It is repeatedly commented in the manuscript, that the reactivity of the title species is still in its infancy and is waiting to be explored, therefore giving the impression that the main focus of the present work is indeed the reactivity. However, this manuscript is mainly devoted to synthesis/characterization and not to reactivity (only one obvious reaction with a potent electrophile is given). Therefore, I would suggest to either (i) not give so much relevance to reactivity in the text or (ii) significantly expand the reactivity part in the manuscript (for instance, with transition metal complexes or other species).

(2) For completeness, it would be helpful if the authors not only show the main NOCV orbital interactions but the rest of the interactions given by the ETS method (in a similar way as Frenking and co-workers do) and compare them with available data for lighter group 15 species.

(3) The molecular orbital showing the remaining lone-pair in the dicationic species should be provided as well.

(4) The computational level shown in the main text does not match that in the computational details (dispersion corrections are missing). Please correct.

Reviewer #2:

Remarks to the Author:

The crystal structures are generally well done but there are a few amendments which should be made:
1: in the supplementary text, the temperature is quoted as 113 K, however in the CIF temperatures are listed as 273 (for 1,2 & 3) and 293 K (for 5). This anomaly need to be addressed and if necessary, the CIFs should be updated in the CCDC.

2: it would be helpful if additional information on the crystal structure processing was provided in the supplementary information. For example, that two of the structures were pseudomeroheredral twins, plus any constraints and restraints reported.

3: The modelling of the disordered CF₃ groups in both 2 and 3 could be improved by using a 3 position model rather than a two position one. This would significantly improve the quality factors and the overall structure.

Subject to these amendments, I am happy with the crystallographic data.

Reviewer #3:

Remarks to the Author:

This manuscript by the group of Mo describes the preparation of cationic antimony(I) and bismuth(I) species (2 & 3) by means of a bis(silylene) ligand (1). Subsequent treatment of cationic species (2 & 3) with MeOTf provide the corresponding dicationic antimony(III) and bismuth(III) species (4 & 5). Standard analytical methods (NMR, HRMS, EA, IR, etc.) have been used for the elucidation of the molecular structures of new compounds. Some of compounds have been determined by XRD analysis. Authors also employed DFT calculations to understand electronic nature of these cationic species.

The related cationic antimony and bismuth compounds have precedence in the literature as correctly referenced by the authors in most cases (Lichtenberg, Okuda, Roesky, Majumdar, Beckmann, etc.). In these literatures, a large number of two ligands coordinate system which is highly relevant to the presented work. Some works by Venugopal, Norman, Sato, and Coles regarding the di- and monocationic bismuth species are missing.

The employed ligand system based of two amidinate N-heterocyclic silylene units have been well established by Driess group. Incorporation of triazaboradecalin can be considered as a new aspect, however, ligand properties of 1 was not well presented. In terms of electronic nature (donor strength) and structural feature, comparison to literature known bis(silylene) ligand system by Driess (e.g. carborane linker, ferrocene linker, arene linker, etc.) should have been described.

Synthetic part for the isolation of compounds (2-5) has been well conducted and DFT results were clearly presented. Supporting information is also acceptable, but Authors should add their comments on Datablock 5 with Alert level B.

Concerning the impact of the work, no reactivity investigations of these cationic compounds weakened the story of the paper and at least statement on Lewis acidity should have been given. This can be done by both experimentally and computationally. Meanwhile, the lone pair of electrons at cationic antimony/bismuth atom towards Lewis acidic compounds can also be considered to proof the Lewis basic nature (see also references 23, 38, etc.)

In conclusion, while the work is technically sound for the most and the manuscript clearly written, the degree of novelty of the reported result is insufficient for Nature Communications as a current form. Upon addressing some concerns outlined above, the revised manuscript will be suitable for publication in a journal for a more specialized audience.

Reviewer: 1

General Comments:

In their manuscript, Mo and co-workers describe the preparation and complete characterization (including X-ray diffraction analysis) of novel As(I) and Sb(I) cations stabilized by a bidentate bis(silylene) ligand. This study follows the previous achievement by Roesky, Stephan and others on strongly related species and is in line with recent computational studies (mainly by Frenking and coworkers) on group 13-15 analogues of tetrylones. In addition, the reactivity of the title compounds was briefly explored in the reaction with MeOTf which leads to the corresponding dicationic species. DFT calculations were in addition carried out to mainly support the occurrence of two lone-pairs in the cations.

In my opinion, this is a solid piece of work that has been competently and carefully carried out. As the manuscript is in general well-written and the literature up-to-date, I am glad to support its acceptance in NatCommun. The following minor issues might be addressed in a revision:

Response:

We thank the reviewer for the highly positive evaluation of this work and for the comments given below. In the light of the reviewers' comments, we have thoroughly revised the manuscript to improve this manuscript.

Comment 1:

It is repeatedly commented in the manuscript, that the reactivity of the title species is still in its infancy and is waiting to be explored, therefore giving the impression that the main focus of the present work is indeed the reactivity. However, this manuscript is mainly devoted to synthesis/characterization and not to reactivity (only one obvious reaction with a potent electrophile is given). Therefore, I would suggest to either (i) not give so much relevance to reactivity in the text or (ii) significantly expand the reactivity part in the manuscript (for instance, with transition metal complexes or other species).

Response:

We thank the reviewer for the comment. The coordination chemistry of the bis(silylene)-stabilized antimony(I) and bismuth(I) cations (**2** and **3**) towards transition metal has been investigated. The coordination of **2** and **3** towards group 6 metal centers (Cr, Mo) is shown to afford a series of unprecedented ionic antimony and bismuth metal carbonyl complexes $[(\text{TBDSi}_2)\text{Pn} \rightarrow \text{M}(\text{CO})_5][\text{BAr}^{\text{F}}_4]$ (Pn = Sb, M = Cr, **6**; Pn = Sb, M = Mo, **7**; Pn = Bi, M = Cr, **8**; Pn = Bi, M = Mo, **9**). Density functional theory (DFT) calculations were performed to gain insight into the electronic structure, stability and bonding nature of **6** and **8**. These new results have been included in the revised manuscript, which can be found on page 8-11.

Comment 2:

For completeness, it would be helpful if the authors not only show the main NOCV orbital interactions but the rest of the interactions given by the ETS method (in a similar way as Frenking and co-workers do) and compare them with available data for lighter group 15 species.

Response:

We thank the reviewer's comments. Followed by the reviewer's suggestion, we did EDA-NOCV analysis on both compounds **2** and **3**, and add detailed discussions on page 6 and 7. In addition, we also conducted EDA-NOCV calculations on the compounds **6** and **8**, which can be found on page 10 and 11. All the changes have been highlight for clear.

Comment 3:

The molecular orbital showing the remaining lone-pair in the dicationic species should be provided as well.

Response:

We thank the reviewer for the comment. The remaining lone pairs located on Sb and Bi atoms can be found in the HOMO-6 and HOMO-8, respectively (see Fig. 8 (b) and Fig. S35).

Comment 4:

The computational level shown in the main text does not match that in the computational details (dispersion corrections are missing). Please correct.

Response:

We thank the reviewer for the comment. The error has been corrected in the revised text.

Reviewer: 2

General Comments:

The crystal structures are generally well done but there are a few amendments which should be made.

Response:

We thank the reviewer for the highly positive evaluation of this work and for the comments given below. In the light of the reviewers' comments, we have thoroughly revised the cif files.

Comment 1:

in the supplementary text, the temperature is quoted as 113 K, however in the CIF temperatures are listed as 273 (for 1,2 & 3) and 293 K (for 5). This anomaly needs to be addressed and if necessary, the CIFs should be updated in the CCDC.

Response:

We thank the reviewer for the comment. The X-ray crystallographic data for compounds **1**, **2**, **3**, **4**, and **5** were collected at 113(2) K. The CIF temperatures have been revised and CIFs have been updated in the CCDC.

Comment 2:

it would be helpful if additional information on the crystal structure processing was provided in the supplementary information. For example, that two of the structures were pseudomerohedral twins, plus any constraints and restraints reported.

Response:

We thank the reviewer's comments. Followed by the reviewer's suggestion, we provide additional information about the crystal structure processing in the revised supplementary information.

Comment 3:

The modelling of the disordered CF₃ groups in both **2** and **3** could be improved by using a 3 position model rather than a two position one. This would significantly improve the quality factors and the overall structure.

Subject to these amendments, I am happy with the crystallographic data.

Response:

We thank the reviewer for the comment. In order to improve the quality factors and the overall structures of **2** and **3**, data for single crystals **2** and **3** have been reprocessed and the R-factors have been decreased.

Reviewer: 3

General Comments:

This manuscript by the group of Mo describes the preparation of cationic antimony(I) and bismuth(I) species (**2** & **3**) by means of a bis(silylene) ligand (**1**). Subsequent treatment of cationic species (**2** & **3**) with MeOTf provide the corresponding dicationic antimony(III) and bismuth(III) species (**4** & **5**). Standard analytical methods (NMR, HRMS, EA, IR, etc.) have been used for the elucidation of the molecular structures of new compounds. Some of compounds have been determined by XRD analysis. Authors also employed DFT calculations to understand electronic nature of these cationic species.

Response:

We thank the reviewer for the highly positive evaluation of this work and for the comments given below. In the light of the reviewers' comments, we have thoroughly revised the manuscript to improve this manuscript.

Comment 1:

The related cationic antimony and bismuth compounds have precedence in the literature as correctly referenced by the authors in most cases (Lichtenberg, Okuda, Roesky, Majumdar, Beckmann, etc.). In these literatures, a large number of two ligands coordinate system which is highly relevant to the presented work. Some works by Venugopal, Norman, Sato, and Coles regarding the di- and monocationic bismuth species are missing.

Response:

We thank the reviewer for the comment. As suggested by the reviewer, the works by Venugopal, Norman, Sato, and Coles regarding the di- and monocationic bismuth species have been cited in ref. 17-22.

Comment 2:

The employed ligand system based of two amidinate *N*-heterocyclic silylene units have been well established by Driess group. Incorporation of triazaboradecalin can be considered as a new aspect, however, ligand properties of **1** was not well presented. In terms of electronic nature (donor strength) and structural feature, comparison to literature known bis(silylene) ligand system by Driess (e.g. carborane linker, ferrocene linker, arene linker, etc.) should have been described.

Response:

We thank the reviewer for the comment. To enable a better understanding of the electronic property and structural feature of bis(silylene) ligand **1**, we have now rewritten the paragraph regarding preparation and characterization of **1**. Its comparison to literature known bis(silylene) ligands reported by Driess have been described in the revised manuscript.

Comment 3:

Synthetic part for the isolation of compounds (2-5) has been well conducted and DFT results were clearly presented. Supporting information is also acceptable, but Authors should add their comments on Datablock 5 with Alert level B.

Response:

We thank the reviewer for the careful examination. According to the reviewer's suggestion, we have added the comments on Datablock 5 in the supporting information.

For compound **5**:

Alert level B

PLAT090_ALERT_3_B Poor Data / Parameter Ratio (Zmax > 18) 5.54 Note

Response: A large number of parameters are increased due to large number of disordered groups in the structure, resulting in a decrease in the data/parameter ratio.

Comment 4:

Concerning the impact of the work, no reactivity investigations of these cationic compounds weakened the story of the paper and at least statement on Lewis acidity should have been given. This can be done by both experimentally and computationally. Meanwhile, the lone pair of electrons at cationic antimony/bismuth atom towards Lewis acidic compounds can also be considered to proof the Lewis basic nature (see also references 23, 38, etc.).

Response:

We thank the reviewer for the comment. To enhance the significance of this work, the coordination chemistry of the bis(silylene)-stabilized antimony(I) and bismuth(I) cations (**2** and **3**) towards transition metal has been investigated. The coordination of **2** and **3** towards group 6 metal centers (Cr, Mo) affords a series of unprecedented ionic antimony and bismuth metal carbonyl complexes [(TBDSi₂)Pn→M(CO)₅][BAR^F₄] (Pn = Sb, M = Cr, **6**; Pn = Sb, M = Mo, **7**; Pn = Bi, M = Cr, **8**; Pn = Bi, M = Mo, **9**). Density functional theory (DFT) calculations were performed to gain insight into the electronic structure and bonding nature of **6** and **8**. These new results have been included in the revised manuscript.

In order to evaluate the Lewis acidity of compounds **2-5**, we calculated the fluoride ion affinity (FIA), and hydride ion affinity (HIA) of compounds **2-5** which is detailed in SI (Fig. S9). The calculated FIA and HIA of **2** (114.2 and 113.0 kcal/mol, respectively), (**3** (115.3 and 113.7 kcal/mol, respectively), **4** (192.8 and 201.4 kcal/mol, respectively) and **5** (190.2 and 254.2 kcal/mol, respectively) are much lower than that of SbF₅ (FIA: 474.9 kcal/mol) and B(C₆F₅)₃ (HIA: 546.5 kcal/mol). The Lewis acidity of **2-5** was further estimated by Gutmann–Beckett method using Et₃PO as an internal standard. When OPET₃ was mixed with one equivalent of **2**, **3**, **4** or **5** in CD₂Cl₂, the peak shifted to δ 50.8, 51.3, 54.0, and 55.2 ppm, respectively, indicating a weak coordination of Et₃P=O to them. These studies show that **4** and **5** are weak Lewis acid despite their dicationic character, which might be due to the strong coordination of two silylene moieties to the Sn and Bi centers.

Table S9. DFT calculations of Gutmann–Beckett, fluoride ion affinity (FIA), and hydride ion affinity (HIA) analyses of compounds **2-5** at the BP86+D3(BJ)/def2-TZVPP level. Energy values are given in kcal/mol.

	FIA	HIA
2	114.2	113.0
3	115.3	113.7
4	192.8	201.4
5	190.2	254.2
SbF ₅	474.9	/
B(C ₆ F ₅) ₃	/	546.5

Table S10. ^{31}P NMR Chemical Shifts (parts per million) and Acceptor Numbers (ANs) of Compounds **2-5** Determined with the Gutmann–Beckett Method with 1 equiv. of OPEt_3 as a Lewis Base.

entry	compound	δ $^{31}\text{P}\{\text{H}\}$ [ppm]	AN
1	2	50.8	22
2	3	51.3	23
3	4	54.0	29
4	5	55.2	31

Comment 5:

In conclusion, while the work is technically sound for the most and the manuscript clearly written, the degree of novelty of the reported result is insufficient for Nature Communications as a current form. Upon addressing some concerns outlined above, the revised manuscript will be suitable for publication in a journal for a more specialized audience.

Response:

We thank the reviewer for the in-depth evaluation and helpful comments on this work. To enhance the significance of this work, we have further studied the coordination chemistry of the bis(silylene)-stabilized antimony(I) and bismuth(I) cations (**2** and **3**) towards transition metals. Heavier group 15 (antimony and bismuth) chemistry is mainly dominated by compounds with the Sb and Bi elements in oxidation state three. Isolation of low-valent cationic antimony(I) and bismuth(I) compounds is very challenging due to their intrinsic kinetic lability, which facilitates decomposition into ‘free’ ligands and elemental Sb and Bi. The key to extension of this chemistry, which is still in its infancy, is to search for new suitable strong σ donor ligands. This work demonstrates the utility of strongly electron-donating bis(NHSi) ligand in stabilization of highly reactive cationic antimony(I) and bismuth(I) complexes, which could serve as a precursor for the preparation of novel methyl antimony(III) and bismuth(III) dications. The coordination of **2** and **3** towards group 6 metals (Cr, Mo) gives unprecedented ionic antimony and bismuth metal carbonyl complexes. We believe that this paper will be of broad general interest, not only to those working in the highly topical field of low valent main group chemistry, but also to colleagues working in the broader fields of coordination chemistry.

We would like to take this chance to thank you and the reviewers very much for editing and reviewing this manuscript. Your valuable comments have made a much better presentation of this paper. I trust these revisions fairly address the points raised by the reviewers and sincerely hope that this revised manuscript is now acceptable for publication. Thank you!

Reviewers' Comments:

Reviewer #1:

Remarks to the Author:

In their revision, the authors have addressed the issues I commented in my previous report in a satisfactory manner. I particularly thank the authors for their efforts toward the synthesis and characterization of the pentacarbonyl-transition metal derivatives and the additional ETS-NOCV calculations.

Regarding the ETS-NOCV calculations, which are convincing, I would suggest to include in the main manuscript the calculations for the Sb-cationic species 3 (Table 1) and the corresponding Cr(CO)₅ complex (Table 2) to enable a rapid comparison with their Bi counterparts.

In addition, the calculations in Table S7 should be removed because the alternative fragmentations (using charged Cr(CO)₅) make no sense at all (the corresponding EDA-NOCV values agree with that). Similarly, the use of triplets and excited singlets (Tables S4 and S6) makes no chemical sense considering the proven singlet-state nature of the title compounds.

Moreover, the literature regarding the ADF basis-set used (TZ2P) and ZORA is missing in the supporting information. There is also a mistake when stating that they used the BP86/def2-SVP geometries (in principle, they were computed at the BP86-D3(BJ)/def2-TZVPP level --- please confirm and correct this point).

The PBE/PBE1 is the keyword used in Gaussian to refer to the PBE0 functional (please correct this in the computational details).

Minor point: on page 8 "There is only one reports..." should read "There is only one report..."

Reviewer #2:

Remarks to the Author:

I am happy that my concerns regarding the crystallographic processing have been addressed. I am happy for the paper to be published.

Reviewer #3:

Remarks to the Author:

In the revised version, authors addressed points raised by reviewers.

The description on the comparison to known bis(NHSi) pincer ligands are well outlined including the ²⁹Si NMR and Si...Si distance. Here are some literatures that can be added to the list as other known other spacer ligands for bis(NHSi) system in order to highlight the novelty of the finding by authors:

Chem. Sci. 2022, 13, 8634.

J. Am. Chem. Soc. 2017, 139, 13499

Organometallics 2014, 33, 6885.

Angew. Chem. Int. Ed. 2012, 51, 3691.

J. Am. Chem. Soc. 2010, 132, 15890.

Kudos to authors for detail computational analysis on these complexes. EDA-NOCV, MOs, NBO/NPA, and also FIA/HIA were well discussed.

Moreover, authors carried out reactivity studies and the demonstrated coordination chemistry of Sb/Bi complexes is also nice addition to the results and a series of Group 6 complexes using Lewis basic Sb/Bi centre. These complexes have been characterized and their structures are elucidated by XRD analysis (Fig 10).

Thus, I am satisfied with the revised version and happy to recommend this paper for publication in Nat Comm after authors address above mentioned minor points.

Reviewer: 1

General Comments:

In their revision, the authors have addressed the issues I commented in my previous report in a satisfactory manner. I particularly thank the authors for their efforts toward the synthesis and characterization of the pentacarbonyl-transition metal derivatives and the additional ETS-NOCV calculations.

Response:

We thank the reviewer for the highly positive evaluation of this work and for the comments given below. In the light of the reviewers' comments, we have thoroughly revised the manuscript to improve this manuscript.

Comment 1:

Regarding the ETS-NOCV calculations, which are convincing, I would suggest to include in the main manuscript the calculations for the Sb-cationic species **3** (Table 1) and the corresponding Cr(CO)₅ complex (Table 2) to enable a rapid comparison with their Bi counterparts.

Response: We thank the reviewer's comments and suggestions. Followed by the reviewer's suggestion, we have included the EDA-NOCV results of Sb-cationic species **3** and the corresponding Cr(CO)₅ complex in Table 1 and Table 2, respectively.

Comment 2:

In addition, the calculations in Table S7 should be removed because the alternative fragmentations (using charged Cr(CO)₅) make no sense at all (the corresponding EDA-NOCV values agree with that). Similarly, the use of triplets and excited singlets (Tables S4 and S6) makes no chemical sense considering the proven singlet-state nature of the title compounds.

Response: Followed by the reviewer's comments, the mentioned results have been deleted in the revised version.

Comment 3:

Moreover, the literature regarding the ADF basis-set used (TZ2P) and ZORA is missing in the supporting information. There is also a mistake when stating that they used the BP86/def2-SVP geometries (in principle, they were computed at the BP86-D3(BJ)/def2-TZVPP level --- please confirm and correct this point).

Response: We thank the reviewer's careful revision. The relevant literatures have been added in the revised version. On the other hand, we checked and confirmed that the geometry optimizations were conducted at the BP86-D3(BJ)/def2-TZVPP level, and thus changed the writing in the computational details section accordingly: "The EDA-NOCV calculations were

carried out at the BP86-D3(BJ)/TZ2P-Zora level^[16,17] by using the optimized geometries obtained at the BP86/def2-TZVPP level." All changes have been highlighted for clear.

Comment 4:

The PBEPBE1 is the keyword used in Gaussian to refer to the PBE0 functional (please correct this in the computational details).

Response:

We thank the reviewer for the comment. The error has been corrected in the revised supporting information.

Comment 5:

Minor point: on page 8 "There is only one reports..." should read "There is only one report..."

Response:

We thank the reviewer for the comment. The error has been corrected in the revised text.

Reviewer: 2

General Comments:

I am happy that my concerns regarding the crystallographic processing have been addressed. I am happy for the paper to be published.

Response:

We thank the reviewer for the highly positive evaluation of this work.

Reviewer: 3

General Comments:

In the revised version, authors addressed points raised by reviewers.

Kudos to authors for detail computational analysis on these complexes. EDA-NOCV, MOs, NBO/NPA, and also FIA/HIA were well discussed.

Moreover, authors carried out reactivity studies and the demonstrated coordination chemistry of Sb/Bi complexes is also nice addition to the results and a series of Group 6 complexes using Lewis basic Sb/Bi centre. These complexes have been characterized and their structures are elucidated by XRD analysis (Fig 10).

Thus, I am satisfied with the revised version and happy to recommend this paper for publication in Nat Comm after authors address above mentioned minor points.

Response:

We thank the reviewer for the highly positive evaluation of this work and for the comments given below. In the light of the reviewers' comments, we have thoroughly revised the manuscript to improve this manuscript.

Comment 1:

The description on the comparison to known bis(NHSi) pincer ligands are well outlined including the ^{29}Si NMR and Si...Si distance. Here are some literatures that can be added to the list as other known other spacer ligands for bis(NHSi) system in order to highlight the novelty of the finding by authors:

Chem. Sci. 2022, 13, 8634.

J. Am. Chem. Soc. 2017, 139, 13499

Organometallics 2014, 33, 6885.

Angew. Chem. Int. Ed. 2012, 51, 3691.

J. Am. Chem. Soc. 2010, 132, 15890.

Response:

We thank the reviewer for the comment. As suggested by the reviewer, the works regarding the other spacer ligands for bis(NHSi) systems have been cited in ref. 57,60-64.

We would like to take this chance to thank you and the reviewers very much for editing and reviewing this manuscript. Your valuable comments have made a much better presentation of this paper. I trust these revisions fairly address the points raised by the reviewers and sincerely hope that this revised manuscript is now acceptable for publication. Thank you!